Integrated transcriptome meta-analysis of pancreatic ductal adenocarcinoma and matched adjacent pancreatic tissues

Atay Sevcan sevcan.atay@ege.edu.tr
Department of Medical Biochemistry, Ege University Faculty of Medicine , Izmir , Turkey
Karakülah Gökhan
Electronic publication date: 2020 Oct 27
Publication date: 2020
Volume: 8
Electronic Location ID: e10141
Received 2020 Jul 27; Accepted 2020 Sep 19
Copyright: © 2020 Atay
Copyright year: 2020
Copyright holder: Atay
License: This is an open access article distributed under the terms of the Creative Commons Attribution License, which permits unrestricted use, distribution, reproduction and adaptation in any medium and for any purpose provided that it is properly attributed. For attribution, the original author(s), title, publication source (PeerJ) and either DOI or URL of the article must be cited.
License URL: https://creativecommons.org/licenses/by/4.0/

Keywords: Pancreatic ductal adenocarcinoma, Microarray, Gene expression, Gene expression omnibus, Biomarker

Funding: The author received no funding for this work.

==============================
A comprehensive meta-analysis of publicly available gene expression microarray data obtained from human-derived pancreatic ductal adenocarcinoma (PDAC) tissues and their histologically matched adjacent tissue samples was performed to provide diagnostic and prognostic biomarkers, and molecular targets for PDAC. An integrative meta-analysis of four submissions (GSE62452, GSE15471, GSE62165, and GSE56560) containing 105 eligible tumor-adjacent tissue pairs revealed 344 differentially over-expressed and 168 repressed genes in PDAC compared to the adjacent-to-tumor samples. The validation analysis using TCGA combined GTEx data confirmed 98.24% of the identified up-regulated and 73.88% of the down-regulated protein-coding genes in PDAC. Pathway enrichment analysis showed that “ECM-receptor interaction”, “PI3K-Akt signaling pathway”, and “focal adhesion” are the most enriched KEGG pathways in PDAC. Protein-protein interaction analysis identified FN1, TIMP1, and MSLN as the most highly ranked hub genes among the DEGs. Transcription factor enrichment analysis revealed that TCF7, CTNNB1, SMAD3, and JUN are significantly activated in PDAC, while SMAD7 is inhibited. The prognostic significance of the identified and validated differentially expressed genes in PDAC was evaluated via survival analysis of TCGA Pan-Cancer pancreatic ductal adenocarcinoma data. The identified candidate prognostic biomarkers were then validated in four external validation datasets (GSE21501, GSE50827, GSE57495, and GSE71729) to further improve reliability. A total of 28 up-regulated genes were found to be significantly correlated with worse overall survival in patients with PDAC. Twenty-one of the identified prognostic genes (ITGB6, LAMC2, KRT7, SERPINB5, IGF2BP3, IL1RN, MPZL2, SFTA2, MET, LAMA3, ARNTL2, SLC2A1, LAMB3, COL17A1, EPSTI1, IL1RAP, AK4, ANXA2, S100A16, KRT19, and GPRC5A) were also found to be significantly correlated with the pathological stages of the disease. The results of this study provided promising prognostic biomarkers that have the potential to differentiate PDAC from both healthy and adjacent-to-tumor pancreatic tissues. Several novel dysregulated genes merit further study as potentially promising candidates for the development of more effective treatment strategies for PDAC.

Introduction

Pancreatic cancer has the highest mortality rate of all solid tumors and ranks fourth on the list of cancer-related causes of death in the world, albeit it represents only 3% of newly diagnosed tumors (Siegel, Miller & Jemal, 2018). According to the American Cancer Society, 57,600 people will be diagnosed with pancreatic cancer in 2020, and 47,050 deaths will be attributed to pancreatic cancer in the United States, reflecting the fatal nature of the disease. It is expected to become the 2nd leading cause of cancer-related death by 2030, surpassing breast, colorectal, and prostate cancer. (Rahib et al., 2014). The most common form of pancreatic cancer is pancreatic ductal adenocarcinoma (PDAC), which accounts for more than 90% of all cases (He et al., 2018).

Despite improvements in survival rates observed in most cancer types, advancements in the treatment of pancreatic cancer have remained steady for more than 40 years, as evidenced by incidence and mortality rates (Siegel, Miller & Jemal, 2018; Wu et al., 2018). The overall 5-year survival rate for patients with pancreatic cancer remains less than 8% and 1-year survival of around 18% when all stages are combined (Saad et al., 2018; Siegel, Miller & Jemal, 2018). Lack of distinctive symptoms in the early stages of the disease, specific risk factors, and an effective screening process eventually causes delayed diagnosis (Maitra & Hruban, 2008; Weledji et al., 2016). Since traditional chemotherapy has limited benefits on survival (Conroy et al., 2011), the only treatment option offering a chance for cure remains as curative surgery for which only 10–20% of patients are considered eligible. However, the majority of patients (50–60%) present with metastatic disease at the time of diagnosis (Gillen et al., 2010), and thus cannot benefit from curative surgery, improving median overall survival to 11–23 months and 5-year overall survival rates to 15–20% (La Torre et al., 2014). Unfortunately, 60% of the patients experience local and systemic relapse within the first 12 months after curative resection (La Torre et al., 2012), and more than 80% of the patients die of the disease due to local recurrence or distant metastasis (Gillen et al., 2010). Together with delayed diagnosis, underlying causes for its exceptionally dismal prognosis also include poor efficacy of treatment modalities such as adjuncts to surgery, undetected micro-metastases, and the development of resistance to chemotherapy (Amrutkar & Gladhaug, 2017). Thus, there is an urgent need for the development of novel and more effective targeted therapies capable of improving survival in patients with pancreatic cancer.

Over the last few decades, various studies have used high-throughput transcriptome profiling to expand our understanding of the underlying molecular mechanisms of pancreatic cancer and to discover novel diagnostic biomarkers and therapeutic targets (Campagna et al., 2008; Tan et al., 2003). An integrative meta-analysis of the transcriptome data allows researchers to combine results from several studies to increase sample size, thereby statistical power and consistency (Ramasamy et al., 2008). Since the first meta-analysis was published in 2005 (Grutzmann et al., 2005), several meta-analyses were conducted to unravel molecular and clinical subtypes of pancreatic cancer (Zhao, Zhao & Yan, 2018), to reveal the genes involved in the prognosis of the disease (Goonesekere et al., 2014; Goonesekere et al., 2018; Haider et al., 2014; Zheng et al., 2018) or to identify novel diagnostic biomarkers (Irigoyen et al., 2018).

My approach to revealing novel deregulated molecular mechanisms underlying PDAC, which differentiates this study from others, is to eliminate the potential influences of clinical, demographic and environmental factors on transcriptome profiles by including only microarray data obtained from human-derived pancreatic ductal adenocarcinoma tissues and their histologically matched adjacent-to-tumor tissue samples in this study. As a result of a careful and detailed examination of the microarray data meeting the stringent inclusion criteria of this study, the present work provides not only new insights into the molecular mechanisms underlying PDAC but also suggests novel biomarkers that may serve as promising indicators of prognosis and diagnosis for PDAC.

Materials and Methods

Selection of microarray datasets

NCBI Gene Expression Omnibus (http://www.ncbi.nlm.nih.gov/geo/) and ArrayExpress Archive of Functional Genomics Data were systematically searched for eligible datasets using the keyword “pancreatic ductal adenocarcinoma”. The inclusion criteria were: (i) gene expression microarray data, (ii) human-derived pancreatic ductal adenocarcinoma tissues and matched adjacent non-tumor tissue samples. When tumor-adjacent tissue pairs were not specified clearly in the overall design of the study or the sample description, the authors of the relevant paper were consulted for the confirmation of sample pairs. Non-confirmed or unspecified sample pairs were excluded from the meta-analysis.

Generation of gene expression matrix files and evaluation of data quality

All data processing and integration procedures were performed using ExAtlas, which is an online software tool for meta-analysis and visualization of gene expression data (Sharov, Schlessinger & Ko, 2015). Briefly, the datasets that have been selected for the meta-analysis were uploaded to ExAtlas. Unpaired samples were removed from the sample files, and then the gene expression matrix file was generated from each dataset separately. All the extracted data had been normalized using the RMA algorithm. Individual sample quality was evaluated by checking the correlation of log10-transformed expression level with other data for a set of pre-selected housekeeping genes and the level of the global standard deviation. Samples, where the correlation of expression of housekeeping genes in the range from 0.5 to 0.95, and the level of standard deviation from the global mean for each set of genes grouped by the average expression is less than 0.3 were considered of good quality. Samples of low quality were removed from the datasets.

Standard meta-analysis

In the Pairwise comparison section of ExAtlas (Sharov, Schlessinger & Ko, 2015), one of the tumor gene expression profiles was added as a sample for examination, and its adjacent non-tumor tissue pair was added for baseline control. Then, the meta-analysis section was used to add more gene expression profile pairs. The random-effects method (DerSimonian & Laird, 1986), which takes into account the variance of heterogeneity between studies, was used to perform the meta-analysis. False discovery rate (FDR) is less than 0.05 and the change of gene expression is ≥2-fold were considered significant. The analysis was performed for each gene symbol, and the effects were presented as combined fold changes and combined log-ratios (log10).

External validation of the identified DEGs

The external validation was done using GEPIA Database (http://gepia.cancer-pku.cn/index.html) (Tang et al., 2017) by comparing transcriptomic data from the TCGA PAAD (pancreatic adenocarcinoma), and the TCGA normal and GTEx data. For genes, whose probes are not found in the validation dataset, Logsdon pancreas dataset (available at the Oncomine Database; Rhodes et al., 2004) including data from pancreatic ductal adenocarcinoma and healthy pancreatic tissues was used. The external validation in a study including PDAC and matched adjacent tissues could not be performed because all studies that passed the inclusion criteria were included in the meta-analysis.

Transcription factor binding site enrichment analysis

TFactS Database (http://www.tfacts.org) (Essaghir & Demoulin, 2012; Essaghir et al., 2010) was used as a tool to predict which transcription factors (TF) are regulated, inhibited, or activated based on the list of DEGs. Briefly, the up-and down-regulated genes were uploaded to the TFact database, and the DEGs were then compared with the sign-sensitive catalog of validated target genes of TFs. Transcription factors whose target genes show a significant overlap with the DEGs were reported. The value of <0.05 for all four indexes (P-value, q-value, E-value, and FDR) in the TFacts Database was considered statistically significant.

Finding prognostic genes for PDAC

Kaplan Meier Plotter (http://kmplot.com/analysis/), an open-access database that provides log-rank P-value and Hazard ratio with 95% confidence intervals for Kaplan–Meier analysis of the correlation between mRNA expression level and patient overall survival (Nagy et al., 2018), was employed to predict the prognostic importance of the DEGs detected in this study. The TCGA Pan-Cancer pancreatic ductal adenocarcinoma cohort was used for the analysis. Auto-select best cutoff value was used to split the patients in survival analysis. P ≤ 0.05 and FDR ≤ 0.05 were used as the cutoff for significance. Then, four independent external validation datasets (GSE21501, GSE50827, GSE57495, and GSE71729) including clinical and transcriptomic data from pancreatic ductal adenocarcinoma patients were analyzed using the PROGgenev2 Prognostic Database (Goswami & Nakshatri, 2014) to evaluate the prognostic relevance of the identified candidate prognostic genes by Km-Plotter. The patient cohorts were divided into two equal groups based on median expression for each gene. P ≤ 0.05 was accepted as statistically significant.

The correlation between mRNA expression of the identified prognostic genes and pathological stages of the disease was evaluated in TCGA PAAD data using the GEPIA Database. P < 0.05 was accepted as statistically significant.

Protein-protein interaction analysis

Search Tool for the Retrieval of Interacting Genes (STRING, version 11.0, https://string-db.org/) is an online database designated to evaluate physical and functional associations of proteins (Szklarczyk et al., 2014). STRING app in Cytoscape Software (version 3.6.1) was used to detect the interactions among the DEGs with a confidence score cut-off >0.9. Then, the protein-protein interaction (PPI) network was constructed using Cytoscape Software (Shannon et al., 2003). The Molecular Complex Detection plug-in (MCODE) was used to identify clusters in the PPI network with the parameters false degree cutoﬀ 2 and K-Core 2 (Bader & Hogue, 2003). Finally, the Gene Ontology and KEGG pathway enrichment analyzes of the DEGs in the clusters with MCODE score >5 were performed with the STRING Enrichment app in Cytoscape Software by retrieving functional enrichment for selected clusters only.

Gene ontology and pathway enrichment analysis of the DEGs

Gene Ontology Enrichment Analysis was performed using the Functional Enrichment Analysis Tool (FunRich) (Pathan et al., 2015) to identify the biological processes, molecular functions, and cellular components that are shared by the up-or down-regulated genes separately. The Gene Ontology Database (The Gene Ontology Consortium, 2017; Ashburner et al., 2000; The Gene Ontology Consortium, 2018) was selected as a background for the analysis, and a P-value of <0.05 was used as the cut-off for significance.

The list of differentially up-and down-regulated genes in PDAC was explored for functionally enriched pathways using a WEB-based GEne SeT AnaLysis Toolkit (WebGestalt 2017) (Wang et al., 2017) with a cut-off criterion of FDR < 0.05. Enrichment categories were selected as Kyoto Encyclopedia for Genes and Genomes (KEGG) (Kanehisa & Goto, 2000), Reactome Pathway Knowledgebase (Fabregat et al., 2018), PANTHER (Thomas et al., 2003), and Wikipathways (Kelder et al., 2012). The category size was calculated based on the number of overlapping genes between the annotated genes in the category and the ranked gene list for the GSEA method. Categories with sizes smaller than 15 and greater than 1,000 were removed during the analysis.

Results

Microarray datasets

The workflow and summary of the results of the presented meta-analysis is shown in Fig. 1. A systematic search of the studies was carried out up to January 2019. The search with the keyword Pancreatic Ductal Adenocarcinoma’ resulted in 18 microarray gene expression datasets which had been submitted only to the ArrayExpress Archive of Functional Genomics Data. Among these, none of the datasets was found to contain adjacent normal tissue as a control group and therefore these datasets were not included in the meta-analysis. Gene Expression Omnibus Datasets were searched with the term “pancreatic ductal adenocarcinoma” and the search results were filtered by selecting organism as homo sapiens, study type as expression profiling by array and attribute name as tissue. As a result, 54 studies were found in the GEO Database. These studies were carefully evaluated and eight studies (GSE62452, GSE15471, GSE62165, GSE56560, GSE60646, GSE55643, GSE18670, and GSE58561) which contain data from 144 PDAC tissues and matched adjacent non-tumor tissue samples were found to meet the inclusion criteria. After exclusion of non-confirmed sample pairs and samples with low quality, 105 PDAC and matched adjacent non-tumor tissue pairs from four datasets (GSE62452, GSE15471, GSE62165, and GSE56560) were decided to be eligible for the meta-analysis (Table 1).

Figure 1 The workflow of the meta-analysis and summary of the results.

Table 1 Eligible public datasets used in the meta-analysis.

Public datasets	Array platform	Number of sample pairs*	PMID	
GSE62452	[HuGene-1_0-st] Affymetrix Human Gene 1.0 ST Array	59	27197190	
GSE15471	[HG-U133_Plus_2] Affymetrix Human Genome U133 Plus 2.0 Array	35	19260470	
GSE62165	[HG-U219] Affymetrix Human Genome U219 Array	9	27520560	
GSE56560	[HuEx-1_0-st] Affymetrix Human Exon 1.0 ST Array	2	25587357	
Note:

* The number of sample pairs after exclusion of non-confirmed sample pairs and samples of low quality.

Differentially expressed genes in PDAC

The meta-analysis identified 344 (342 protein-coding genes) differentially over-expressed and 168 (157 protein-coding genes) repressed genes in PDAC compared to the matched adjacent non-tumor samples (Table S1). A list of the top 20 differentially expressed genes (FDR < 0.05) with at least a 2-fold differential expression between groups is shown in Table 2. Along with the genes reported to be associated with pancreatic cancer in previous studies, the results of the meta-analysis revealed candidate over-expressed genes for PDAC, many of which have previously received little or no attention, such as KYNU (kynureninase), ITGBL1 (integrin beta-like 1), and ADGRF1 (adhesion G protein-coupled receptor).

Table 2 Ranked list of the top 10 up and down-regulated genes in pancreatic ductal adenocarcinoma.

Gene symbol	Gene name	Logratio combined	Fold change	P-value	FDR	
POSTN	periostin, osteoblast specific factor	1.0004	10.009	0	0	
CEACAM5	carcinoembryonic antigen-related cell adhesion molecule 5	0.8855	7.683	0	0	
SLC6A14	solute carrier family 6 (amino acid transporter), member 14	0.8611	7.263	0	0	
CEACAM6	carcinoembryonic antigen-related cell adhesion molecule 6 (non-specific cross reacting antigen)	0.8452	7.002	0	0	
SULF1	sulfatase 1	0.8347	6.835	0	0	
LAMC2	laminin subunit gamma 2	0.8279	6.728	0	0	
FN1	fibronectin 1	0.8083	6.432	0	0	
COL11A1	collagen, type XI, alpha 1	0.7918	6.191	0	0	
INHBA	inhibin beta A	0.7713	5.907	0	0	
VCAN	versican	0.7644	5.813	0	0	
ALB	albumin	-0.8658	−7.342	0	0	
SERPINI2	serpin peptidase inhibitor, clade I (pancpin), member 2	−0.783	−6.068	8.88E−16	4.96E−14	
PNLIPRP1	pancreatic lipase-related protein 1	−0.7683	−5.866	9.40E−12	4.71E−10	
ERP27	endoplasmic reticulum protein 27	−0.7367	−5.454	6.66E−16	3.75E−14	
PNLIPRP2	pancreatic lipase-related protein 2	−0.7359	−5.444	3.66E−12	1.85E−10	
CTRL	chymotrypsin-like	−0.7199	−5.247	3.55E−15	1.94E−13	
PDIA2	protein disulfide isomerase family A member 2	−0.7025	−5.041	0	0	
GP2	glycoprotein 2 (zymogen granule membrane)	−0.7005	−5.018	3.06E−11	1.51E−09	
CELA2B	chymotrypsin like elastase family member 2B	−0.685	−4.842	7.62E−12	3.83E−10	
IAPP	islet amyloid polypeptide	−0.6768	−4.751	8.88E−16	4.96E−14	

The differentially expressed genes identified in PDAC was validated in TCGA combined GTEx data containing data from PDAC tissues and pancreatic tissues from healthy individuals. This confirmation was also aimed at determining whether the DEGs identified could also distinguish PDAC tissues from healthy pancreatic tissues. Gene expression analysis of TCGA combined GTEx data validated the up-regulation of 336 and the down-regulation of 116 protein-coding genes in PDAC (P ≤ 0.001, Fold Change ≥2). Additionally, the elevated expression of AK4 was validated in the Logsdon Pancreas Dataset in Oncomine Database due to the absence of the probes for this gene in TCGA combined GTEx data (P = 1.16E−5). The identified DEGs and the results of the validation analysis are shown in Table S1.

Taken together, the expression of a significant portion of the identified DEGs was consistent with the gene expression profile from TCGA, implying that the identified genes may have the potential to discriminate PDAC from both adjacent and healthy pancreatic tissues.

Activated and inhibited transcription factors in PDAC

To predict which transcription factors (TF) are activated or inhibited to regulate the transcription of the identified DEGs, TFs whose target genes show a significant overlap with the DEGs were identified using TFactS Database. Table S2 shows the results of the TF enrichment analysis. A total of five transcription factors were found to reach statistical significance (P-value, q-value, E-value, and FDR < 0.05). Among these, Transcription Factor 7 (TCF7), Catenin Beta 1 (CTNNB1), Smad Family Member 3 (SMAD3) and Jun Proto-Oncogene (JUN) were found to be significantly activated in PDAC while Smad Family Member 7 (SMAD7) was detected as significantly inhibited.

Prognostic values of the identified DEGs in PDAC

The prognostic values of the identified DEGs genes were evaluated by analyzing the TCGA Pan-Cancer pancreatic ductal adenocarcinoma data and survival outcomes of patients included in the Km-Plotter. Sixty-one up-regulated and one down-regulated genes (P ≤ 0.05 and FDR ≤ 0.05) were found to be significantly associated with the overall survival rate of patients with PDAC. TCGA combined GTEx data validated the up- and down regulation of the identified candidate prognostic genes in PDAC. Subsequently, four external datasets (GSE21501, GSE50827, GSE57495, and GSE71729) were analyzed to validate the prognostic significance of the identified 62 candidate genes for PDAC. The candidate prognostic gene list identified by Km-Plotter and the results of the validation analysis are shown in Table S3.

Genes whose prognostic potentials were confirmed in one of the four validation sets were included in the “prognostic gene list” and their predicted prognostic values were ranked based on (i) validation status (validated in 1-2-3-4 dataset(s) =validation status 1-2-3-4), (ii) the average of p-values, and (iii) the average of hazard ratios. None of the genes could reach validation status 4. However, two genes, LAMC2 and ITGB6, were found to be significantly correlated with worse overall survival in three of the four validation datasets. SERPINB5, KRT7 TGM2, IGF2BP3, INPP4B, IL1RN, DCBLD2, and MPZL2 were validated in two datasets. Three genes, TMPRSS4, SEMA3C, and MGLL, were associated with increased overall survival in one of the validation datasets, therefore excluded from the prognostic gene list. Gene expression data of CDK1, PKM, AK4, AREG were not available in some validation datasets. While AK4 and PKM, which were found to be correlated with overall survival in one of the four external datasets, were included in the prognostic gene list, CDK1 and MGLL were excluded from the list due to insufficient gene expression data.

Consequently, a total of 28 up-regulated genes were individually validated in at least one of the four external validation datasets. The ranked prognostic gene list for PDAC and the results of the Kaplan–Meier survival analysis in five datasets are shown in Table 3. Kaplan–Meier survival plots for the identified prognostic genes are shown in Fig. 2.

Table 3 The ranked prognostic gene list for PDAC and the results of the Kaplan–Meier survival analysis in five datasets.

	Gene symbol			Validation datasets	Val. Stat.			
		TCGA
Pan-Cancer PDAC	GSE21501	GSE50827	GSE57495	GSE71729		Average	
		HR	P	HR	P	HR	P	HR	P	HR	P		P	HR	
1	ITGB6	2.59	4.90E−06	1.39	0.001	NS	1.54	0.002	1.30	0.01	3	3.38E−03	1.7050	
2	LAMC2	3.06	1.10E−04	1.53	0.003	NS	1.33	0.03	1.24	0.01	3	1.06E−02	1.7900	
3	KRT7	3.12	7.20E−05	1.29	0.02	NS	1.46	0.004	NS	2	9.16E−03	1.9567	
4	SERPINB5	2.89	5.40E−06	1.37	0.01	NS	1.23	0.03	NS	2	1.18E−02	1.8300	
5	IGF2BP3	3.55	1.00E−05	1.17	0.04	NS	1.28	0.01	NS	2	1.40E−02	2.0000	
6	DCBLD2	2.19	2.00E−04	NS	NS	1.57	0.00	1.29	0.04	2	1.54E−02	1.6833	
7	TGM2	2.14	3.00E−04	NS	1.42	0.04	1.48	0.01	NS	2	1.68E−02	1.6800	
8	INPP4B	3.07	1.00E−04	1.37	0.03	NS	1.24	0.03	NS	2	1.94E−02	1.8933	
9	IL1RN	2.39	7.00E−04	1.29	0.02	NS	1.26	0.05	NS	2	2.29E−02	1.6467	
10	MPZL2	2.30	1.00E−04	NS	1.47	0.05	1.36	0.05	NS	2	3.14E−02	1.7100	
11	SFTA2	2.39	2.00E−04	NS	NS	1.37	0.00	NS	2	5.50E−04	1.8800	
12	MET	2.79	1.20E−07	NS	NS	1.68	0.001	NS	2	7.00E−04	2.2350	
12	LAMA3	3.86	3.60E−06	1.50	0.002	NS	NS	1.57	0.001	2	9.68E−04	2.3100	
14	DHRS9	2.14	3.00E−04	NS	NS	1.31	0.004	NS	1	1.90E−03	1.7250	
15	FRMD6	2.33	3.00E−04	NS	NS	1.65	0.01	NS	1	3.15E−03	1.9900	
16	ARNTL2	2.51	7.40E−06	1.47	0.01	NS	NS	NS	1	3.15E−03	1.9900	
17	PKM	2.52	1.00E−05	1.88	0.01	N/A	N/A	N/A	1	3.51E−03	2.2000	
18	SLC2A1	3.73	4.40E−05	NS	NS	1.33	0.01	NS	1	4.02E−03	2.5300	
19	LAMB3	2.18	3.00E−04	NS	NS	NS	1.23	0.01	1	6.65E−03	1.7050	
20	COL17A1	2.19	2.00E−04	NS	NS	1.20	0.03	NS	1	1.36E−02	1.6950	
21	EPSTI1	2.22	2.00E−04	NS	NS	NS	1.45	0.03	1	1.51E−02	1.8350	
22	IL1RAP	2.51	1.00E−04	NS	NS	NS	1.56	0.03	1	1.71E−02	2.0350	
23	AK4	2.26	7.40E−05	NS	N/A	1.30	0.04	N/A	1	1.75E−02	1.7800	
24	ANXA2	2.50	8.40E−06	1.61	0.04	NS	NS	NS	1	1.85E−02	2.0550	
25	S100A16	2.16	2.00E−04	1.40	0.04	NS	NS	NS	1	1.86E−02	1.7800	
26	KRT19	3.23	7.90E−05	NS	NS	NS	1.22	0.04	1	1.90E−02	2.2250	
27	GPR87	3.37	4.60E−06	NS	NS	1.15	0.04	NS	1	2.15E−02	2.2600	
28	GPRC5A	2.64	4.60E−06	NS	NS	1.26	0.05	NS	1	2.40E−02	1.9500	
Note:

HR, hazard ratio; NS, nonsignificant (p > 0.05); N/A, not available; P, P value; Val. Stat, validation status; PDAC, Pancreatic ductal adenocarcinoma.

Figure 2 Kaplan–Meier survival plots for the identified up-regulated genes in PDAC (A–BB).

Survival plots were created using Km-Plotter. Kaplan–Meier survival plots are shown only for genes whose elevated expressions were significantly associated with the overall survival rate of patients in TCGA data and whose prognostic values were validated in at least one of the external validation datasets (GSE21501, GSE250827, GSE57495, and GSE71729).

Further analysis of TCGA pancreatic ductal adenocarcinoma data in GEPIA showed that, twenty-one of the identified prognostic genes (ITGB6, LAMC2, KRT7, SERPINB5, IGF2BP3, IL1RN, MPZL2, SFTA2, MET, LAMA3, ARNTL2, SLC2A1, LAMB3, COL17A1, EPSTI1, IL1RAP, AK4, ANXA2, S100A16, KRT19, and GPRC5A) were also correlated with the pathological stages of the disease, underlying their prognostic value for PDAC (Fig. 3).

Figure 3 The identified prognostic genes whose mRNA expressions were found to be correlated with the pathological tumor stages in patients with PDAC (A–U).

Violin plots were created using GEPIA based on the TCGA PAAD dataset. F-value indicates the statistical value of F test; Pr (>F) indicates P-value. P < 0.05 was accepted as statistically significant.

Protein-protein interaction (PPI) network

The PPI analysis was performed to evaluate the physical and functional associations of proteins encoded by the identified differentially expressed genes in PDAC. The PPI network was constructed by Cytoscape based on the PPI correlations from the STRING database. PPIs among the DEGs with a confidence score cut-off >0.9 were selected to construct the PPI network (Fig. 4). Then the degree of each node in the network was calculated by using the NetworkAnalyzer tool of Cytoscape to identify hub proteins in the PPI network. The degree of a node is the number of edges connected to the node, and it has been stated that nodes with higher degrees -which correspond to hub proteins in the PPI network-, play an essential role in the organization of the PPI network, therefore might be more crucial and relevant than non-hub genes (Vallabhajosyula et al., 2009). In this study, nodes with degrees >15 are considered to indicate “hub proteins” and are presented in Table 4. Next, the PPI network was analyzed by MCODE to identify clusters in the network. Each clustered protein group was then analyzed to predict its biological function in PDAC.

Figure 4 The protein-protein interaction (PPI) network analysis of differentially expressed genes in PDAC.

The network was constructed by Cytoscape based on the PPI correlations from the STRING database. The clusters in the network was identified using MCODE. A total of nine clusters with MCODE score >5 were marked and named with different colors in the network.

Table 4 The list of the identified hub protein-coding genes in PDAC.

Gene symbol	Node degree	Gene name	
FN1	25	Fibronectin type III domain containing	
TIMP1	23	Tissue inhibitor of metalloproteinases 1	
MSLN	22	Pre-pro-megakaryocyte-potentiating factor	
FBN1	20	Fibrillin 1	
ALB	20	Serum albumin	
F5	20	Coagulation factor V (proaccelerin, labile factor)	
SERPINA1	20	Serpin peptidase inhibitor, clade A (alpha-1 antiproteinase, antitrypsin), member 1	
COL1A2	20	Collagen alpha-2(I) chain	
IGFBP3	19	Insulin-like growth factor binding protein 3	
COL3A1	19	Collagen alpha-1(III) chain	
COL1A1	19	Collagen alpha-1(I) chain	
ITGA2	18	Integrin, alpha 2 (CD49B, alpha 2 subunit of VLA-2 receptor)	
COL17A1	18	180 kDa bullous pemphigoid antigen 2	
TNC	18	Glioma-associated-extracellular matrix antigen	
SPP1	18	Secreted phosphoprotein 1	
VCAN	17	Chondroitin sulfate proteoglycan core protein 2	
MATN3	17	Matrilin 3	
IGFBP5	17	Insulin-like growth factor binding protein 5	
EGF	16	Pro-epidermal growth factor	
GNB4	16	Guanine nucleotide binding protein (G protein)	
LTBP1	16	Latent transforming growth factor beta binding protein 1	
COL4A2	16	Collagen alpha-2(IV) chain	
LGALS1	16	Lectin, galactoside-binding, soluble, 1	
APOL1	16	Apolipoprotein L, 1	
COL11A1	16	Collagen alpha-1(XI) chain	
ANXA1	16	Phospholipase A2 inhibitory protein	
CP	16	Ceruloplasmin (ferroxidase)	

Among the 27 hub proteins, ITGA2 and COL17A1 were found to be associated with unfavorable prognosis for PDAC based on the TCGA Pan-Cancer pancreatic ductal adenocarcinoma dataset in Km-Plotter (P ≤ 0.05 and FDR ≤ 0.05). However, this finding could not be validated in the four external datasets, except the significant correlation between the up-regulation of COL17A1 and decreased overall survival found in GSE57495. Moreover, Fibronectin 1 (FN1), tissue inhibitor of metalloproteinases 1 (TIMP1), and fibrillin 1 (MSLN) constituted the top three proteins with degrees exceeding 20. All these three proteins were members of CLUSTER1, which includes proteins whose functions mostly associated with extracellular matrix signaling pathways and structural organization, underlying the importance of extracellular dynamics in the pathogenesis of PDAC. The most significantly up-regulated gene in this study, POSTN, was found to be located close to the collagen sub-cluster included in CLUSTER1, emphasizing their close interactions. CLUSTER1 also included up-regulated genes with unknown importance in pancreatic cancer, such as MATN3, SERPINA1, and IGFBP5.

CLUSTER3 covered known prognostic biomarkers such as TOP2A (Tsiambas et al., 2007), CDK1 (Piao et al., 2019), and MKI67 (Striefler et al., 2016). This cluster was surrounded by genes (CENPK, ANLN, ECT2, and FAM83D) related to the cell cycle, but it appears that they were not included in the cluster due to the stringent confidence score cut-off criteria of the analysis. Amongst, the function of FAM83D in PDAC is not yet known. However, there are previous reports that describe this protein as a probable proto-oncogene that regulates cell proliferation, growth, migration, and epithelial to mesenchymal transition (Wang et al., 2015; Wang et al., 2013), suggesting that it may have similar functions in PDAC.

Furthermore, CLUSTER2 and 4, were mapped to the chemokine signaling pathway and type I interferon signaling pathway, respectively, and located close to each other in the PPI network. Although the importance of chemokine signaling in PDAC has been previously reported (Geismann et al., 2019), the exact role of cellular innate antiviral response in the pathogenesis of PDAC remains elusive. In this study, OAS1, OAS2, and RSAD2, which play critical roles in cellular innate antiviral responses induced by type I and type II interferon, were found to be up-regulated in PDAC. Additionally, Kaplan–Meier survival analysis of the TCGA data using Km-Plotter indicated an association of high OAS1 expression with the low survival rate of PDAC patients, however, could not be validated in other four external validation datasets. The roles of these proteins in PDAC are still unknown, and further research is needed for the evaluation of their potential as diagnostic or prognostic biomarkers for PDAC.

While CLUSTER6 shows the known interleukin-laminin-EGF crosstalk in the focal adhesion pathway in PDAC, CLUSTER9 indicated a novel down-regulated gene in PDAC, P2RX1, a ligand-gated ion channel with relatively high calcium permeability, that may be linked to apoptosis by increasing the intracellular concentration of calcium in the presence of ATP (Uniprot Database, by similarity). CLUSTER8 included cell surface antigens such as CD109, CD66e, and CD90. However, as shown in the PPI network, the interaction between these proteins and the proteins that have roles in the regulation of glycolysis such as PKM, PFKP, and ENO2 was intriguing and may indicate stemness associated interactions in PDAC. The other clusters were significantly mapped to those specific KEGG pathways: “Mucin-Type O-glycan Biosynthesis” (CLUSTER5) and “Estrogen Signaling Pathway” (CLUSTER7), indicating once more the high expression of keratins and mucin-type O-glycans in pancreatic tumors. The proteins assigned to the pathways are shown in Table S4.

Gene ontology analysis of the DEGs

Gene ontology analysis of the DEGs was performed to identify enriched biological processes, molecular functions and cellular locations associated with differential gene expression in PDAC. As shown in Fig. 5A, the analysis identified nine significantly overrepresented individual categories of GO Molecular Function for the up-regulated genes including “extracellular matrix structural constituents” (P < 0.001), “collagen binding” (P < 0.001) and “integrin-binding” (P < 0.001). “Extracellular matrix organization” and “cell adhesion” were the most significantly enriched biological processes (P < 0.001, Fig. 5B), while most of the proteins encoded by the up-regulated genes were found to be located in the “collagen-containing extracellular matrix” (P < 0.001, Fig. 5C).

Figure 5 Gene Ontology analysis of the differentially expressed genes in PDAC.

Enriched molecular functions (A and D), biological processes (B and E) and cellular locations (C and F) associated with the differential gene expression in PDAC were shown. Analyses were performed using FunRich.

For the down-regulated genes, “serine-type endopeptidase activity” was the only GO molecular function category that reached statistical significance (P = 0.008, Fig. 5D). Three GO Biological Processes were found to be significantly enriched for the down-regulated genes: “proteolysis” (P = 0.001), “cellular zinc ion homeostasis” (P = 0.002), and “cellular response to copper ion” (P = 0.042), (Fig. 5E). Moreover, most of the over-represented GO cellular locations were associated with extracellular space for the down-regulated genes (Fig. 5F).

Pathway enrichment analysis of the DEGs

Significantly dysregulated pathways in PDAC were identified by using four different pathway databases in WebGestalt (Table S5). The obtained results from the pathway enrichment analysis of the identified DEGs underlined the importance of the crosstalk between tumor cells and extracellular matrix, including integrin and PI3K-Akt-mTOR signaling pathways, in PDAC pathogenesis. Pathways related to metabolism and pancreatic secretion were identified as negative related categories by KEGG and Reactome Pathway Knowledgebase.

Discussion

The lack of an effective treatment option for pancreatic cancer emphasizes an absolute need for expanding our knowledge of the etiology and molecular mechanisms of the disease to discover novel drug targets. In the current study, using stringent inclusion and exclusion criteria examining patient selection and microarray quality assessment, an integrative meta-analysis of transcriptome data from four studies was conducted to suggest novel multifunctional biomarkers as well as therapeutic targets for pancreatic ductal adenocarcinoma.

Comparison of tumor and adjacent tissues has advantages such as providing more reliable results due to the elimination of the variations between individuals and anatomical locations from which samples are taken. This approach allowed the identification of potential biomarkers for PDAC that may serve in the molecular evaluation of the surgical margin, which provides a more sensitive and precise assessment of the risk of cancer recurrence than solely by histopathologic examination. Although histologically normal adjacent-to-tumor tissues are generally accepted as healthy controls, there are also studies reporting that, at the molecular level, these tissues are distinct from both healthy and tumor tissues and represent an intermediate state (Aran et al., 2017; Russi et al., 2019). Therefore, the potential of the defined DEGs in PDAC to be diagnostic biomarkers was evaluated in an external validation dataset including PDAC and healthy pancreatic tissues (TCGA combined GTEx data). The validation analysis verified 98.24% of the identified up-regulated and 73.88% of the identified down-regulated protein-coding genes in PDAC, suggesting that these genes may serve as promising diagnostic biomarkers that differentiate PDAC from both healthy and adjacent-to-tumor pancreatic tissues.

Kinase inhibitors constitute a significant portion of chemotherapeutic agents that are in clinical use today. The results of this meta-analysis revealed that the expression of fifteen kinase-encoding genes that have the potential to be therapeutic targets in PDAC, is higher than both healthy and adjacent pancreatic tissues. Additional survival analysis on these individual genes revealed that higher expression of Diacylglycerol Kinase η (DGKH) and adenylate kinase 4 (AK4) was associated with worse survival probabilities in at least two of the five external datasets. DGKH plays a crucial role in promoting cell growth and activates the RAS/Raf/MEK/ERK signaling pathway induced by EGF (Yasuda et al., 2009). AK4 is a critical mitochondrial enzyme that participates in maintaining the homeostasis of cellular nucleotides and plays a role in controlling cellular ATP levels by regulating AMPK signaling (Lanning et al., 2014). However, the functional roles and the clinicopathological significance of AK4 and DGKH in pancreatic cancer have never been investigated. Here, the results of this study showed that increased AK4 and DGKH expressions are independently correlated with a decreased overall survival rate of patients with PDAC. It is also noteworthy that, DGKH was found to be a prognostic marker in three external validation datasets (GSE21501, GSE50827, and GSE57495), indicating its prognostic power as a biomarker (Fig. S1). These results suggest that AK4 and DGKH may be potential therapeutic targets in devising a treatment for patients with pancreatic cancer and have the potential to be prognostic and diagnostic biomarkers for PDAC.

In the context of the prognostic significance of the identified DEGs, genes were analyzed using Km-Plotter Database to see whether the differentially expressed genes identified in this study had a significant effect on the survival of patients with PDAC, as survival data were not available for some GEO datasets used in the meta-analysis. To increase the reliability, the identified candidate genes whose expressions significantly correlated with the overall survival rate of patients were further validated using four distinct GEO Datasets. The results of this analysis revealed a total of twenty-eight up-regulated genes in PDAC compared to both adjacent and normal pancreatic tissues that might have the potential to be prognostic and diagnostic biomarkers for PDAC. Notably, violin plots of gene expression by pathological stages based on the TCGA PAAD data showed that twenty-one of the identified prognostic genes (ITGB6, LAMC2, KRT7, SERPINB5, IGF2BP3, IL1RN, MPZL2, SFTA2, MET, LAMA3, ARNTL2, SLC2A1, LAMB3, COL17A1, EPSTI1, IL1RAP, AK4, ANXA2, S100A16, KRT19, and GPRC5A) also significantly correlated with pathological stages of the disease, indicating that these genes may play crucial roles in the tumorigenesis of PDAC. As expected, some of these identified genes have been reported to be associated with poor overall survival of patients with pancreatic cancer (Cheng et al., 2019; Hu et al., 2019; Jahny et al., 2017; Pei, Yin & Liu, 2018; Reader et al., 2019; Schaeffer et al., 2010; Takahashi et al., 2019; Wu et al., 2019; Yao et al., 2016; Zhang et al., 2019). However, the functional and clinicopathological significance of IL1RN, MPZL2, SFTA2, EPSTI1, IL1RAP, AK4 and S100A16 in PDAC have not been reported. Additionally, to the best of my knowledge, a correlation between the elevated expressions of ITGB6, LAMC2, LAMA3, ARNTL2, LAMB3, COL17A1, and GPRC5A and pathological stages of the disease has yet been stated. Taken together, the results of this analysis revealed that these genes individually might have high diagnostic and prognostic values, as well as the potential to be therapeutic targets for PDAC, prompting further study.

Moreover, to predict which transcription factors (TF) are altered to regulate the transcription of the identified DEGs, a TF enrichment analysis was conducted. Among the five TFs identified, activation of SMAD3 and inhibition of SMAD7 may together indicate a possible activation of TGF-beta signaling, which is known to play a dual role as both pro-tumorigenic and tumor-suppressive in pancreatic cancer, depending on tumor stage and microenvironment (Shen et al., 2017). Additionally, Transcription factor 7 (TCF7) and Catenin Beta-1 (CTNNB1), key proteins in the Wnt Signaling Pathway, were predicted to be activated in PDAC. Although there are scientific reports on the prognostic values of SMAD3 (Yamazaki et al., 2014) and CTNNB1 (Zhang et al., 2016) in pancreatic cancer, the precise functions of Jun Oncogene (JUN) and TCF-7 in the pathogenesis of pancreatic cancer are still largely unknown and require further understanding and research.

Further analysis showed that most of the proteins encoded by the identified up/down-regulated genes are components of the extracellular matrix, which is in line with the fact that PDAC has a characteristically abundant desmoplastic stroma (Cannon et al., 2018). Moreover, the gene ontology analysis revealed that the identified up-regulated genes in PDAC generally encode extracellular matrix (ECM) structural elements together with the proteins with collagen and integrin-binding properties. These genes were mapped significantly to the biological processes called ECM organization and cell adhesion, which may promote survival, proliferation, and metastasis of PDAC that result in an aggressive disease phenotype as suggested elsewhere (Weniger, Honselmann & Liss, 2018).

Significantly enriched pathways for up-regulated genes were found to include ECM organization and receptor interaction, focal adhesion-PI3K-Akt-mTOR signaling pathway, and integrin signaling pathway, which have well-known associations with pancreatic cancer (Ebrahimi et al., 2017; Topalovski & Brekken, 2016). These pathways have been known to be activated by various types of cellular stimuli and interact with each other in a variety of complex ways (Hastings et al., 2019; Wu et al., 2016), emphasizing the intricate molecular pathogenesis of pancreatic cancer. In this study, the genes assigned to the mentioned pathways included FN1 (fibronectin) gene, a member of the top three hub proteins identified by the PPI network analysis. Fibronectin, an abundant stromal protein in PDAC, has been known to drive metastatic spread, angiogenesis, and chemoresistance of PDAC by mediating FAK dependent activation of the PI3K/AKT/mTOR pathway or RAS/RAF/MEK pathway (Topalovski & Brekken, 2016). However, in this study FN1 expression was not found to be a potential prognostic factor for overall survival in patients with PDAC. Moreover, LAMB3, another protein assigned to these pathways, has also been shown by a recent study to mediate apoptotic, proliferative, invasive, and metastatic behaviors in pancreatic cancer by regulating the PI3K/AKT signaling pathway (Zhang et al., 2019). The other proteins in these pathways were members of the families of interleukin (ITGA2, ITGB6), laminin (LAMC2), and collagen (COL1A1/2, COL6A3), together with cartilage oligomeric matrix protein (COMP), whose role in pancreatic cancer has not yet been clarified. Amongst, increased expressions of LAMB3 and ITGB6 were significantly associated with poor prognosis of patients with pancreatic adenocarcinoma in this study, underlying the importance of the identified enriched pathways in PDAC.

Regarding the functions of the down-regulated genes in PDAC, the GSEA pathway analysis revealed these genes significantly associated with pancreatic secretion and metabolic pathways. Abnormal pancreatic secretion is known to occur frequently in patients with pancreatic cancer. However, alterations in pancreatic enzyme secretion have not yet been well-characterized in PDAC. In this study, 18 genes associated with pancreatic secretion were found to be down-regulated in PDAC compared to adjacent tissues, including chymotrypsin-like elastase family members (CELA2A/B, CELA3A/B), carboxypeptidases (CPA1/2, CPB1), pancreatic lipase (PNLIP), pancreatic amylase (AMY2A) and a sodium bicarbonate cotransporter; SLC4A4. The causes of abnormal pancreatic secretion other than pancreatic duct obstruction may be a decrease in the number of secretory cells or translational alterations in secretory cells, and whether these alterations contribute to malignancy needs to be investigated.

Altered metabolism and metabolic plasticity have been associated with proliferation, aggressivity, adaptability to changes in the tumor microenvironment, and drug resistance in pancreatic cancer (Biancur & Kimmelman, 2018). In this study, the genes assigned to the suppressed metabolic pathways were mostly enzyme-coding genes associated with amino acid metabolism (ABAT, GATM, GLS2, ANPEP, PSAT1, GPT2, and GAMT). The other down-regulated genes related to various metabolic pathways in PDAC were ADH1B, CTH, EPHX2, AOX1, ACADL, IMPA2, ACAT1, ACSM3, UGT2A3, and CBS. The roles of these genes in pancreatic cancer are not yet known; therefore, further studies addressing this issue may reveal therapeutic, diagnostic, or prognostic values of altered metabolism in PDAC. Moreover, Regucalcin (RGN), which is a highly conserved calcium-binding protein, was also assigned to the metabolic pathways in this study. Overexpression of Regucalcin has been demonstrated to suppress proliferation, cell death, and migration in an in vitro model of pancreatic cancer in a previous study (Yamaguchi et al., 2016). Therefore, the identified down-regulation of RGN in PDAC may emphasize the tumor suppressor property of this protein.

In conclusion, this study provided a list of multifunctional biomarkers that have the potential to distinguish PDAC from both adjacent-to-tumor tissues and healthy pancreatic tissues, as well as correlated with overall survival rate of patients and the pathological stages of the disease. Future investigations are necessary to additionally validate the combined prognostic and diagnostic value of the identified biomarkers in this study. The functional significance of some of these identified dysregulated genes in the pathogenesis of PDAC is not yet known and deserves further investigation. Moreover, the results of this study provided insights into the molecular basis of the difference between PDAC and adjacent-to-tumor tissues which may be useful in the histopathological examination of PDAC and in the development of more effective targeted therapies.

Supplemental Information

Supplemental Information 1 DGKH was found to be a prognostic marker for PDAC in three external GEO datasets (GSE21501, GSE50827, and GSE57495).

Kaplan-Meier plots for DGKH created using PROGgeneV2. The patient cohorts were divided into two equal groups based on median expression for DGKH. P<0.05 was accepted as statistically significant.

Click here for additional data file.

Supplemental Information 2 The identified up- and down-regulated genes in PDAC.

NS: p≥ 0.05, N/A; not available.

Click here for additional data file.

Supplemental Information 3 The predicted transcription factors activated or inhibited in PDAC based on the lists of up-regulated and down-regulated genes resulted from the meta-analysis.

Light colored rows: activated TFs, dark colored row: inhibited TF.

Click here for additional data file.

Supplemental Information 4 The results of the Kaplan-Meier survival analysis of the identified DEGs in five external datasets.

NS: nonsignificant, p≥ 0.05; N/A: not available, p: p value, HR: Hazard ratio. Underlined HR values indicate an association with good outcome.

Click here for additional data file.

Supplemental Information 5 Enriched categories and corresponding enriched genes for MCODE clusters in the protein-protein interaction network.

Click here for additional data file.

Supplemental Information 6 The statistics of the enriched pathways and the leading edge genes.

Lines with white color represent the positively related pathways while lines with gray color represent the negatively related pathways. Size is the number of overlapping genes between categories and the differentially expressed gene list. P<0.05 and FDR<0.05 are accepted as statistically significant. P and/or FDR values <2.220446e-16 were shown as 0e+00. NS: Normalized enrichment score, ES: Enrichment Score, LEN: Leading Edge Number.

Click here for additional data file.

Abbreviations List

TCGA The Cancer Genome Atlas

PDAC Pancreatic Ductal Adenocarcinoma

GTEx The Genotype-Tissue Expression

ECM Extracellular matrix

GEO Gene Expression Omnibus

RMA Robust Multi-array Average

GEPIA Gene Expression Profiling Interactive Analysis

PAAD Pancreatic Adenocarcinoma

TF Transcription Factor

KEGG Kyoto Encyclopedia of Genes and Genomes

DEGs Differentially expressed genes

PPI Protein-protein interaction

GO Gene Ontology

EGF Epidermal Growth Factor

Additional Information and Declarations

Competing Interests

Author Contributions

Data Availability

The author declares that they have no competing interests.

Sevcan Atay conceived and designed the experiments, performed the experiments, analyzed the data, prepared figures and/or tables, authored or reviewed drafts of the paper, and approved the final draft.

The following information was supplied regarding data availability:

The data is available at NCBI GEO: GSE62452, GSE15471, GSE62165, GSE56560, GSE21501, GSE50827, GSE57495, GSE71729.

The TCGA-PAAD and TCGA-Pan-cancer raw data is available at UCSC Zena (available at https://xenabrowser.net/datapages/).

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
