# Peer review of "Integrated transcriptome meta-analysis of pancreatic ductal adenocarcinoma and matched adjacent pancreatic tissues"

_PeerJ, doi:10.7717/peerj.10141_

## Round 0.1 · original submission · Minor Revisions

Both reviewers indicated some minor points. To improve the content and readability of the manuscript, I would like to invite the author to revise her manuscript.

·

Basic reporting

-

Experimental design

-

Validity of the findings

-

Additional comments

The current manuscript by Sevcan Atay presents a transcriptome meta-analysis of pancreatic cancer tissues. GEO, TCGA combined with GTEx data sets and others were utilized to identify pancreatic cancer specific genes. Pathway enrichment analysis was carried out as well as survival analysis using TCGA Pan-Cancer data. Genes specific for pancreatic cancer, correlating with survival of pancreatic cancer patients were identified.
This is an interesting analysis that is solely based on publicly available data and data analysis tools. The novelty of the findings is rather limited. The utilized methods are sound and valid. However, there are further concerns that should be addressed:
It is difficult to follow, which datasets were used as training and validation cohorts. Obviously, these datasets are all different, therefore validation is not trivial. Could the author please comment?
Using an inclusion criterion such as “high quality gene expression microarray data” does not necessarily mean that these data are of higher quality than other published microarray data.
Sample pairs can make sense but need better explanation. In pancreatic cancer, most areas next to caner contain chronic pancreatitis like changes and not ‘normal’ pancreatic tissue. This should be discussed.
The work provides “a list of multifunctional biomarkers that have the potential to distinguish PDAC from both adjacent-to-tumor tissues and healthy pancreatic tissues”. This statement is true, and the methodology is up-to-date. However, there are numerous studies with similar aims and results. Validation and functional characterization remain most important.

Reviewer 2 ·

Basic reporting

No comment

Experimental design

No comment

Validity of the findings

No comment

Additional comments

This is a very nice bioinformatics work and a well written manuscript. Using many public databases and analysis tool that all of them are freely available Dr Atay comes to some interesting findings on biomarkers for PDAC. I only wonder if a more detailed analysis that could link biomarkers to staging of PDAC would be able. This could help to identify biomarkers for the early detection of this deadly cancer which is may be the biggest clinical problem.
Also it would be very helpful I believe for the reader Dr Atay to include a paragraph where all abbreviations will be explained (there are a lot!) and a table with all the databases and tools used for the meta-analysis.

---

## Round 0.2 · accepted · Accept

The author successfully implemented the criticisms and comments raised by the reviewers during the revision period. I congratulate the author for her work.

·

Basic reporting

-

Experimental design

-

Validity of the findings

-

Additional comments

The authors have satisfactorily answered most of my and the other reviewers’ questions. I believe that this is an interesting and valid manuscript that has been strengthened by the additions and changes. I have no further comments.